# Not only a Problem of Fatigue and Sleepiness: Changes in Psychomotor Performance in Italian Nurses across 8-h Rapidly Rotating Shifts

**DOI:** 10.3390/jcm8010047

**Published:** 2019-01-05

**Authors:** Marco Di Muzio, Flaminia Reda, Giulia Diella, Emanuele Di Simone, Luana Novelli, Aurora D’Atri, Annamaria Giannini, Luigi De Gennaro

**Affiliations:** 1Department of Clinical and Molecular Medicine, St. Andrea Hospital, University of Rome “Sapienza”, I-00118 Rome, Italy; marco.dimuzio@uniroma1.it (M.D.M.); giulia.die@gmail.com (G.D.); 2Department of Psychology, University of Rome “Sapienza”, I-00185 Rome, Italy; flaminia.reda@uniroma1.it (F.R.); luananovelli@yahoo.it (L.N.); aurora.datri@uniroma1.it (A.D.); Annamaria.Giannini@uniroma1.it (A.G.); 3Department of Biomedicine and Prevention, University of Rome Tor Vergata, I-00133 Rome, Italy; emanuele.disimone@uniroma1.it

**Keywords:** sleepiness, shift work, tiredness, vigilance, Psychomotor Vigilance Task (PVT)

## Abstract

Although many studies have detailed the consequences of shift work in nurses concerning health, fatigue, sleepiness, or medical errors, no study has been carried out trying to disentangle the contribution of sleepiness and fatigue associated to shift work from the attentional performance. The aim of this pilot study is (A) to investigate the effects of an 8-h rapidly rotating shift on fatigue and sleepiness among staff nurses and (B) how these factors affect their psychomotor performance. Fourteen nurses were selected for a within-subject cross-sectional study according to this sequence of shifts: morning–afternoon–night, which were compared as function of tiredness, sleepiness, and performance at the Psychomotor Vigilance Task (PVT). Subsequently, a within-subject Analysis of Covariance (ANCOVA) evaluated if the observed differences between shifts persist when the contribution of sleepiness is controlled. Our results clearly indicate that night shifts are associated with significant greater sleepiness and tiredness, and worsened performance at the PVT. As hypothesized, ANCOVA showed that these differences disappear when the contribution of sleepiness is controlled. Results point to a lower psychomotor performance in night compared to day shifts that depends on sleepiness. Hence, interventions to minimize the consequences of the night shift should consider a reduction of sleepiness.

## 1. Introduction

The night shift, with its overnight duty hours, is a prominent feature of health care providers and a necessary part of a 24-h healthcare institution [1,2]. During clinical cares, time pressure and increasing fatigue have additive negative effects, such as in reducing alertness [3] and enhancing the risk for errors in clinical judgment [4], in administration of medications, in decision-making, and new or unfamiliar professional maneuvers that require complex cognitive skills [5,6,7].

Sleep of shift workers is intrinsically associated with a circadian desynchronization disturbed and frequently interrupted to the point that they complain of a reduced amount of sleep in the arc of a month. In fact, ~92% of nurses working on shifts, even at night, report an average of four hours per night of restful sleep within a month [8].

Prolonged work shifts, or night shifts, can therefore lead to a poor quantity/quality of sleep. In fact, homeostatic (i.e., accumulated time awake) and circadian factors negatively affect night shift work, producing a level of fatigue progressively increasing across the duty period [9]. Medication errors and, more in general, performances are mainly compromised by the altered sleep–wake rhythm. In nursing students, this issue leads to a reduction in academic achievement, learning, and behavioral performance [10]. Tiredness, as a consequence of insufficient or poor sleep quality over a prolonged period, can lead to difficulty in attention and concentration, reduced motivation, irritability, misperception, memory lapses, decreased reaction times, loss of empathy, and errors of judgment. In fact, a considerable amount of studies has confirmed that loss of sleep negatively affects cognitive performance, resulting in slower reaction times and poverty in decision-making processes [11,12,13]. Therefore, the decline of performance appears as a relevant issue in night shifts. A study in Italian medical specialists showed also a deterioration of performance in executive functions and concentration-related procedures during the night shifts [14].

The use of the Psychomotor Vigilance Task (PVT), which is considered the most widespread and reliable method for assessing behavioral consequences of sleep deprivation and of excessive sleepiness [15], has recently shown that nurses working with a 12-h shift schedule have greater behavioral deterioration and increased sleepiness during their night shift, and this worsening was accentuated by the end of a 12-h shift [16]. However, other studies have not found differences between 12-h shifts using the PVT, with a dissociation with subjective measures actually showing the expected differences [17]. To the best of our knowledge, a similar investigation has never been carried out in Italian healthcare, which mainly uses 8-h rapidly rotating shifts. Similarly, attempts to disentangle possible different effects of tiredness and sleepiness have been rarely performed.

The aim of our pilot study will be to assess the levels of psychomotor vigilance fatigue and sleepiness across nursing 8-h rapidly rotating work shifts, in order to highlight a potential level of risk associated with a specific work shift. This evaluation will be performed using a within-subject design. Since tiredness and sleepiness may covary with performance at the PVT, we also evaluated if differences in the attentional performance will be still present when partialling the contribution of correlated variables. According to our hypothesis, the differences between different shifts will be not present when the contribution of sleepiness has been taken into account.

## 2. Experimental Section

### 2.1. Subjects

Sixteen nurses (12 females and 4 men) participated in the study, but the final sample consisted of 14 subjects, due to the lacking measures for two participants (i.e., they did not fulfill the condition of three consecutive shifts). Fourteen nurses (10 females and 4 men) between ages of 26 and 47 years (mean age = 36.78, s.d. = 8.90) working with an 8-h rapidly rotating shift in the Emergency Department at Policlinico Umberto I of Rome were selected according to the following criteria.
-Absence of medical conditions and chronic diseases as assessed by a clinical interview;-all the participants have to ensure the performance of shift work with the following sequence: morning (M)—afternoon (A)—night (N). Change of shifts was not allowed for the three days under investigation;-napping was not allowed during the night shift;-all trainees were excluded from the study.

Each participant took part in the project anonymously, in order to guarantee maximum confidentiality on the data collected and signed an informed consent, where the modalities and aims of the protocol were approved by the Ethical Committee of Sapienza University of Rome (# 343/17 on 26 April 2017).

### 2.2. Measures

The Pittsburgh Sleep Quality Index (PQSI) is a self-assessment questionnaire consisting of nine items and provides a reliable, valid, and standardized sleep quality measurement. The PQSI evaluates sleep habits during a period relative to the month before the one of the evaluation. The scale consists of 19 items divided into seven subscales that evaluate the subjective quality of sleep, sleep latency, sleep duration, habitual sleep efficiency, sleep disorders, the use of hypnotic drugs, and disorders during the day. The items are all evaluated on a 0–3 point Likert scale, and in all cases, a score of “0” indicates the absence of difficulty while a score of “3” indicates the presence of serious difficulty. The sum of the scores of the seven components gives the overall score, which has a range between 0 and 21, with “0” indicating the absence of difficulty and “21” serious difficulties in all areas. Scores above five are indicative of the presence of poor sleep quality. We used an Italian validation of this test [18].

The Karolinska Sleepiness Scale (KSS) (Italian adaptation [19]) is a 9-item self-assessment questionnaire and measures the level of sleepiness. Level 1 indicates a state of maximum alertness and level 9 of maximum sleepiness. The subject must indicate the level that best reflects the psychophysical state in the 5 min before the administration of the scale. The KSS score decreases during periods of sustained vigilance, and it is strongly correlated with the time of day.

The Tiredness Symptoms Scale (TSS) (Italian adaptation [20]) is a self-assessment questionnaire, a checklist of 14 physical and emotional symptoms that the patient can experience at the time of evaluation. The questionnaire assesses the physical and emotional symptoms of fatigue at the time of evaluation. Subjects indicated if they had = yes or did not have the symptoms indicated in the list, so the score is obtained by summing the positive responses related to the symptoms. Yes = 1, No = 0.

The Psychomotor Vigilance Task (PVT) is considered a very reliable method for assessing the consequences of sleep deprivation [15]. During the PVT, subjects are placed in front of a computer screen for 5 min, with the task of clicking the left mouse button every time a number appears at irregular intervals. Even a half-second of delay in response may be evidence of lack of sleep, also known as microsleep. In particular, the PVT was set with a random interstimulus interval, not exceeding 100 s.

### 2.3. Procedures

All the participants in the study received the Pittsburgh Sleep Quality Index (PSQI; Italian adaptation [18]), in order to identify the possible presence of sleep disorders. Specifically, the quality, duration, and efficiency of sleep and the impact of somnolence on daytime efficacy were assessed.

Each participant was interviewed at the end of each work shift: morning (7:00–13:30), afternoon (13:30–20:00), and night (20:00–6:30). Their rapidly rotating shift was as follows, day A: Morning, day B: Afternoon, day C: Night, day D: discount, and Day E: Rest. That said, all nurses took part in the project at the end of the shift and for three consecutive days, starting from day A and up to day C. Each subject has been invited to participate in the study at the end of the work shift, for a maximum duration of 15 min. To keep the experiment as controlled as possible, during the sessions the nurses were interviewed and tested in a room without noise and away from any environmental distraction. They were asked to switch off the mobile phones throughout the test, and no entry was allowed to anyone except the experimenter.

KSS, TSS, and PVT were administered in a fixed order. The identical procedural sequence was repeated at the end of each of the three shifts investigated.

### 2.4. Data Analysis

The following dependent variables were considered; TSS and KSS scores and medians of PVT reaction times. These measures were submitted to a repeated measure Analysis of Variance (ANOVA) comparing the three shifts (M vs. A vs. N), and *t*-tests were used for the post-hoc comparisons between the three means. Planned comparisons further compared daytime shifts (morning and afternoon) vs. night shift. The intercorrelations between the dependent variables were assessed by Pearson’s correlations. Finally, an Analysis of Covariance (ANCOVA) compared the three shifts, considering sleepiness or tiredness as covariates if significant correlations will be found between PVT and KSS or TSS, respectively.

## 3. Results

Our sample of nurses reports a quite poor sleep quality as expressed by the PSQI scores (X = 6.36, s.d. = 2.24), and 64.29% of them scored >5 at this test (i.e., the cut-off for PSQI scores).

Figure 1 shows the means and standard deviations of the dependent variables considered in the study. The ANOVA on TSS scores was significant (F_2,26_ = 9.76; *p* = 0.001; Figure 1A). Both M and A shifts were significantly different from N shift (t_13_ = 4.40; *p* = 0.0007 and t_13_ = 3.01; *p* = 0.01, respectively), while M vs. A was not significantly different (t_13_ = 0.48; *p* = 0.64). Not surprisingly, the orthogonal planned comparison confirms that diurnal (M and P) TSS scores were lower than those at N shift (Figure 1).

In the same vein, the ANOVA on the KSS scores was significant (F_2,26_ = 8.34; *p* = 0.002; Figure 1B). Again, M and A shifts were both significantly different from N shift (t_13_ = 3.73; *p* = 0.002 and t_13_ = 2.97; *p* = 0.01, respectively), while M vs. A was not significantly different (t_13_ = 1.07; *p* = 0.31). Hence, also the orthogonal planned comparison confirms that diurnal KSS scores than those at N shift (F_1,13_ = 14.61; *p* = 0.002).

As expected, the ANOVA comparison on the medians of RTs at the PVT was significant (F_2,26_ = 3.63; *p* = 0.04). Both M and A shifts were significantly different from N shift (t_13_ = 2.06; *p* = 0.05 and t_13_ = 2.31; *p* = 0.04, respectively), while M vs. A was not significantly different (t_13_ = 1.00; *p* = 0.33). There was also a significant difference when day and N shifts (F_1,13_ = 5.28; *p* = 0.04).

Although the significant changes across shifts have the same general pattern of differences, only KSS and PVT were significantly correlated in the A shift and showed a statistical tendency in the N shift (r_M_ = 0.14 (*p* = 0.61); r_A_ = 0.77 (*p* = 0.001); r_N_ = 0.44 (*p* = 011)). The intercorrelations between TSS and PVT and between TSS and KSS were never significant.

According to our hypothesis, the ANCOVA on PVT medians and considering KSS as a covariate was not more significant (F_2,24_ = 1.31; *p* = 0.29), while the KSS covariate was significant (Wilk’s Lambda = 0.069, Rao R_9,19_ = 4.37; *p* = 0.003).

## 4. Discussion

To the best of our knowledge, this is the first study investigating the behavioral consequences of 8-h rapidly rotating shifts with the use of the PVT. Our data confirm an increased sleepiness and a greater sense of fatigue at the end of the night shift compared to daytime shifts (i.e., morning and afternoon shifts). More importantly, performance at the PVT also is deteriorated during the night compared to day shifts. This deterioration implies lower sustained-attention and speed with which subjects respond to a visual stimulus. Subjective sleepiness but not tiredness was linearly correlated to the performance at the PVT. When this contribution of sleepiness to psychomotor performance is removed by an ANCOVA design, the three shifts were not significantly different. These findings point to sleepiness as a major candidate to explain the less efficient performance during the night shift. Furthermore, our nurses also report poor sleep quality as expressed by the PSQI scores.

Nurses working at night (i.e., fixed night shift) and nurses working in rotation shifts not only have difficulty staying awake during work but also tend to perform more medical errors [7,21,22,23] compared to those working in day shifts [8,17]. Furthermore, the probability of attentional lapses during the shift or educational activities can also be attributed to the number of excessive duration of shifts [24]. The most original contribution of the current pilot study is however represented by the finding that the statistical differences between shifts disappear when the contribution of sleepiness is partialized, while this does not seem true for tiredness. In other words, it suggests that a relevant role in the behavioral worsening during night shifts may depend on sleepiness more than tiredness. However, our findings are limited to an Emergency Department, and the possibility of significant differences with other workplaces in acute care cannot be disregarded. Although we do not find empirical studies on this specific issue, the question of the generalization to other acute care units is worthy of being investigated, also because the strategy of using planned naps [14,25] to improve alertness and performance cannot be implemented in all workplaces.

One limit of our pilot study is represented by the lack of a direct and objective measure of sleep in the days before the evaluated shifts, and by the fact that the small sample size precluded an acceptable measurement of medical errors. Although underrecognized, medical errors have a significant impact on society in broad terms, since it has been estimated that they are the third biggest cause of death in the US [26].

Among the many unanswered questions in shift design in healthcare that deserve empirical and applicative attention, there is the direction of the rotational patterns. Although some support for the use of forward rotating shift systems in preference to backward rotating shift systems has been reported, at last as far as 8-h shifts [27], a definite conclusion still lacks. For this reason, we will directly assess this difference in a future study with a larger sample of Italian nurses. Another critical issue is the type of rotation (i.e., the rapidly rotating shift). Although this is the most used schedule in Italian healthcare, it is possible that it magnifies sleepiness. A future study should assess the behavioral and attentional consequences of different types of rotation, by comparing rapidly vs. slowly shifts. In fact, a common complaint of shift workers is that their schedule changes too often, precluding to their sleep schedule to adjust after each phase advance [28]. Finally, it should be considered the possibility that different work environments of the units may affect the extent of sleepiness and potential medical errors. This possibility could be empirically assessed by a comparison of samples working in different units.

Concerning developing adequate countermeasures, an Italian study in medical specialists showed how the improvement in executive functions and concentration-related procedures could be achieved by taking a nap during the night shift [14]. However, this strategy seems privileged when longer than 8-h shifts are considered. In our opinion, the core issue is represented by continuous monitoring of sleep–wake schedules in nurses and, in general, in medical personnel. Actigraphy [29] and some innovative sleep apps [30] may be considered as a useful tool for estimating sleep measures and trigger appropriate countermeasures when a sleep disorder or an altered sleep–wake schedule is suspected. Not alternatively, healthcare should consider the possibility to introduce a behavioral measure for monitoring sleepiness during night shifts. Finally, we think that the most crucial change in nursing practice should consider effective treatments for reducing sleepiness associated with night shifts. For this reason, we plan to introduce an exposition to bright light [31] at the start of the night shift. According to our hypothesis, this strategy should significantly reduce sleepiness at the levels of morning and afternoon shifts.

## Figures and Tables

**Figure 1 jcm-08-00047-f001:**
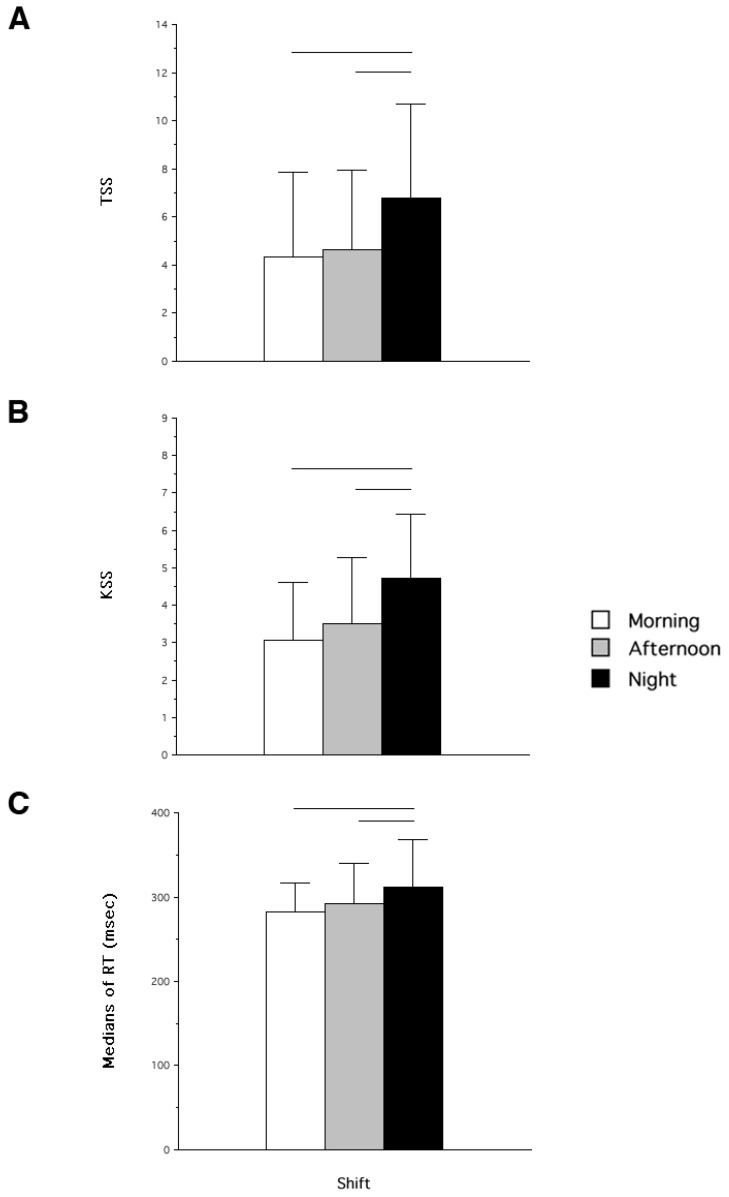
Means and standard deviations of the dependent variables measures in the sample of Italian nurses collected across different rapidly rotating shifts (morning, afternoon, and night). Panel **A** = scores at the Tiredness Symptoms Scale (TSS), panel **B** = scores at the Karolinska Sleepiness Scale (KSS), and panel **C** = medians of Reaction Times (RT) at the Psychomotor Vigilance Task (PVT).

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
