# Peer review of "Not only a Problem of Fatigue and Sleepiness: Changes in Psychomotor Performance in Italian Nurses across 8-h Rapidly Rotating Shifts"

_jcm, 2019, doi:10.3390/jcm8010047_

Round 1

Reviewer 1 Report

This paper is well-written and relatively easy to understand in terms of the rationale for the research, the experimental methods and the analysis.

The authors analysis relative to the effect of sleepiness is aptly demonstrated.

The authors raise some interested issues concerning shift design in healthcare.

As a small pilot study the authors point to the lack of measurement of medical errors.

It is not clear to me what is meant by "the limit of our pilot study is represented by a direct measure  of sleep...."  Does this mean that there was no direct measure of sleep collected?  I would rewrite the sentence if this is the case.

Two additional comments that I would like answered.  Why were rapidly rotating shifts used?  I am under the impression that this type of rotation is not common. Was the effect of sleepiness magnified by the rotating shift design? The study was conducted in an Emergency Department (ED).  Are there situational factors in EDs that set them apart from other units such as ICU, Orthopedic, Medical, Surgical?  If so, what are the implications for the study since the work environments of the units varies.

Author Response

This paper is well-written and relatively easy to understand in terms of the rationale for the research, the experimental methods and the analysis.

The authors analysis relative to the effect of sleepiness is aptly demonstrated.

The authors raise some interested issues concerning shift design in healthcare.

As a small pilot study the authors point to the lack of measurement of medical errors.

R: Thanks for your appreciation and your comments

It is not clear to me what is meant by "the limit of our pilot study is represented by a direct measure  of sleep...."  Does this mean that there was no direct measure of sleep collected?  I would rewrite the sentence if this is the case.

R: According to the reviewer suggestion we have rewritten the sentence.

Two additional comments that I would like answered.  Why were rapidly rotating shifts used?  I am under the impression that this type of rotation is not common. Was the effect of sleepiness magnified by the rotating shift design?

R: The reviewer is right, and we have claryfied  this point. Although this is the most used in Italian healthcare, it is possible that it magnifies sleepiness. A future study should assess behavioral and attentional consequences of different types of rotation, by comparing rapidly vs. slowly rotations.

The study was conducted in an Emergency Department (ED).  Are there situational factors in EDs that set them apart from other units such as ICU, Orthopedic, Medical, Surgical?  If so, what are the implications for the study since the work environments of the units varies.

R: Also in this case the reviewer is right, and different work environments of the units may affect the extent of sleepiness and medical errors. Also this point has been mentioned in our revision.

Reviewer 2 Report

Dear Authors.

I think it is an interesting paper. However, I would like you to comment on the relation to medical errors and the impact that might have on society in broad terms. Since a bad working environment might be costly for society in diverse ways. I think this could be discussed in the discussion part. In the background, I would like to know more about the ED compared to other 24/7 workplaces in acute care. Is this rare or is it common, what have others done?

Thank You

Author Response

I think it is an interesting paper. However, I would like you to comment on the relation to medical errors and the impact that might have on society in broad terms. Since a bad working environment might be costly for society in diverse ways. I think this could be discussed in the discussion part. In the background, I would like to know more about the ED compared to other 24/7 workplaces in acute care. Is this rare or is it common, what have others done?

R: Thanks for your comments. We have now discussed both these issues. Concerning the comparison between ED and other workplaces in acute care there are not -at best of our knowledge- empirical studies on this issue. We have mentioned this question and we have made reference to the strategy of planned naps .